# A Multilevel Intervention Framework for Supporting People Experiencing Homelessness with Pets

**DOI:** 10.3390/ani10101869

**Published:** 2020-10-13

**Authors:** Nick Kerman, Michelle Lem, Mike Witte, Christine Kim, Harmony Rhoades

**Affiliations:** 1School of Psychology, University of Ottawa, Ottawa, ON K1N 6N5, Canada; 2Community Veterinary Outreach, Carp, ON K0A 1L0, Canada; michelle.lem@vetoutreach.org (M.L.); mikewit38@yahoo.ca (M.W.); 3Independent Researcher, Ridgewood, NY 11385, USA; christinehykim@gmail.com; 4USC Suzanne Dworak-Peck School of Social Work, University of South California, Los Angeles, CA 90089, USA; hrhoades@usc.edu

**Keywords:** homelessness, homeless youth, pet ownership, companion animals, One Health, housing policy, service delivery, access to care, Housing First, veterinary medicine

## Abstract

**Simple Summary:**

Approximately one in 10 people experiencing homelessness have pets. Despite pets having psychosocial benefits for their owners, pets can also present challenges for people experiencing homelessness related to meeting their basic needs and obtaining housing. This article proposes a framework of policy, public, and service interventions for improving the health and well-being of pet owners experiencing homelessness. At the policy level, the framework proposes an increase of pet-friendly emergency shelters, access to market rental housing and veterinary medicine, and the use of a Housing First approach. At the public level, educational interventions are needed to improve knowledge and reduce stigma about the relationship between homelessness and pet ownership. At the service delivery level, direct service providers can support pet owners experiencing homelessness by recognizing their strengths, connecting them to community services, being aware of the risks associated with pet loss, providing harm reduction strategies, documenting animals as emotional support animals, and engaging in advocacy. By targeting policies and service approaches that exacerbate the hardships faced by pet owners experiencing homelessness, the framework is a set of deliberate actions to better support this vulnerable group.

**Abstract:**

Approximately one in 10 people experiencing homelessness have pets. Despite the psychosocial benefits derived from pet ownership, systemic and structural barriers can prevent this group from meeting their basic needs and exiting homelessness. A multilevel framework is proposed for improving the health and well-being of pet owners experiencing homelessness. Informed by a One Health approach, the framework identifies interventions at the policy, public, and direct service delivery levels. Policy interventions are proposed to increase the supply of pet-friendly emergency shelters, access to market rental housing and veterinary medicine, and the use of a Housing First approach. At the public level, educational interventions are needed to improve knowledge and reduce stigma about the relationship between homelessness and pet ownership. Direct service providers can support pet owners experiencing homelessness by recognizing their strengths, connecting them to community services, being aware of the risks associated with pet loss, providing harm reduction strategies, documenting animals as emotional support animals, and engaging in advocacy. By targeting policies and service approaches that exacerbate the hardships faced by pet owners experiencing homelessness, the framework is a set of deliberate actions to better support a group that is often overlooked or unaccommodated in efforts to end homelessness.

## 1. Introduction

Homelessness is a complex social problem that refers to people residing in emergency shelters or crisis accommodations, living on the streets or in vehicles, and staying temporarily with friends and family. Estimated rates of homelessness in the United States and Canada—over 567,000 and 35,000 people on any given night, respectively—have remained stagnant or modestly increased in recent years [1,2]. Similar trends have also been found across Europe, with the exception of Finland where there has been a marked reduction in homelessness [3]. For most people, homelessness is experienced as a single episode that is short in length but, for others, homelessness is longer in duration and re-occurring [4,5,6].

The homeless population is highly marginalized and heterogenous. Groups that are overrepresented include people who have physical and mental disabilities, have substance use problems, and are members of racialized groups [1,2,7,8]. Furthermore, people experiencing homelessness are vulnerable to developing medical conditions, including tuberculosis, hepatitis B and C viruses, HIV, skin and foot problems, and oral health problems [9,10,11,12]; being physically and sexually assaulted [13,14]; being arrested [15]; being socially isolated [13,16]; and experiencing stigma and discrimination [17]. People experiencing homelessness are also at-risk of dying younger than the general population, with life expectancies between 8–22 years shorter than those without histories of homelessness [18,19,20]. Given the deleterious and potentially life-threatening consequences of homelessness, it is critical that service and policy interventions meet the diverse needs of the population. 

One group that faces unique challenges is people experiencing homelessness who have pets. Pets are defined as people’s companion animals (these terms are used interchangeably throughout this article) and do not refer to service animals, such as guide dogs, or emotional support animals, which are deemed to have therapeutic benefits to their owners who have a mental illness. The prevalence of pet ownership among the homeless population has been largely overlooked in research. Because of this, estimates have varied considerably, ranging from approximately 5–25% [21,22,23,24,25]. However, given several recent point-in-time counts in the U.S., 10% appears to be a more reliable estimate of the prevalence of pet ownership among the homeless population [25,26]. It is unknown if pet ownership rates differ between homeless youth, single adults, or families.

There are similarities and differences in the characteristics of people experiencing homelessness who own pets and those who do not. In a study of over 4000 homeless adults in Knoxville, Tennessee, pet owners were more likely to be female, be unmarried, be Euro-American, and have previously experienced domestic violence than people who did not own pets [22]. Most pet owners were experiencing homelessness for the first time in their lives, though this was not significantly different from non-pet owners. Youth experiencing homelessness who own pets are also more likely to be female and white [27,28]. In addition, approximately half of homeless youth who have pets are transient, which is significantly higher than those without pets [28]. There are also some preliminary data on the mental health of homeless youth who have pets. Depressive symptoms are reported to be fewer among youth who have pets than those who do not [27,28]. In contrast, pet ownership is not associated with any differences in trauma histories [28]. 

People can develop strong attachments with their pets that yield psychosocial benefits [29,30]. This is also true for people experiencing homelessness who have pets [31,32]. Studies have shown that people experiencing homelessness report that their pets provide a sense of responsibility and are a reason to live, reduce substance use, and seek healthcare [33,34,35,36,37,38]. Moreover, pets are viewed as a stable source of social support and companionship, which is often absent in the lives of people experiencing homelessness [23,28,33,35,36,38]. Relatedly, pets can provide an opportunity to give and receive unconditional love, an experience that may be otherwise difficult to have while living precariously without a home [28,33,34,37]. As boredom is a common issue for people experiencing homelessness [39,40], the responsibilities associated with pet ownership may also buffer against this daily stressor [33]. For some people, particularly youth, pet dogs also offer protection and safety while living on the streets [28,35,37]. As such, pets fulfill basic human needs, including those that are commonly unmet due to homelessness.

Pets can also yield barriers and vulnerabilities that complicate people’s experiences of homelessness. Most notably, many homeless sector services, including emergency shelters, have policies that forbid pets [25,41]. For example, in a recent U.K. study, slightly over 60% of homeless sector services were not accepting of pets [42]. Similarly, nearly half of pet owners experiencing unsheltered homelessness in Los Angeles, California, report that they had been turned away from emergency shelters due to pet-related policies [25]. Non-pet-friendly policies force people to choose between their pets and the services they need. Given the strong attachments to their pets, it is not uncommon for people experiencing homelessness to forgo accessing shelter or healthcare to stay with their animals [25,36,43,44,45]. Similarly, people experiencing homelessness commonly encounter barriers to obtaining housing due to non-pet-friendly policies in rental markets, which can prolong homeless episodes [22,28,43,46]. Other challenges associated with pet ownership include the financial costs of feeding and caring for the animals; access to veterinary care; emotional vulnerability to potential pet loss and its painful consequences; stigmatization and discrimination due to having a pet but not a home; and finding safe and reliable pet care or temporary boarding to access health services or attend vocations [23,28,35,36,47].

Homelessness often forces people to choose between competing sustenance needs for shelter and housing, income, food, and healthcare [48]. Because most health and homeless sector services do not accommodate companion animals, pet owners face additional barriers to addressing their unmet needs. Accordingly, to reduce the disadvantage experienced by pet owners experiencing homelessness, modifications are needed to the approaches and interventions used with this group.

## 2. One Health and Its Implications for Intervention 

The concept of One Health recognizes that the health of humans, animals, and environments are inextricably connected [49]. The model has been largely applied to the study of zoonoses and issues related to food safety [50], though it also has important implications for how human-animal bonds can be leveraged to improve population health. As pets are viewed to be family members in 80% of households in the U.S. [51], the psychological attachment between humans and their companion animals is evident. Moreover, pets have been linked to greater social support and capital, community integration, and physical activity in the general population, which results from more time spent out in one’s neighborhood with pets [52,53,54,55]. Although some of these impacts have not yet been investigated in the homeless population, there is clear evidence that pets can have both positive and negative health effects on their owners [32]. Accordingly, by recognizing this interplay between animal and human health, which is further shaped by living without a home in an often uncertain and unsafe environment, the One Health model enables a more comprehensive understanding of the drivers of health for pet owners experiencing homelessness [56].

Adoption of a One Health approach to care for pet owners experiencing homelessness would represent a change from current support models. As few health and social services accommodate the pets of people experiencing homelessness, the contributions of companion animals to the health and well-being of their owners are either overlooked or de facto viewed to be inconsequential. Accordingly, a One Health approach to care offers an opportunity for validation of pet owners by attending to both them and their companion animals in service delivery. A second shift would be needed in the types of support services available to the homeless population. The support needs of people experiencing homelessness are recognized as diverse, requiring a range of health and social services [57,58,59,60]. However, veterinary care is seldom integrated into service system frameworks for supporting the homeless population. By attending to the connection between human and animal health, a One Health approach would require pet-related supports be accessible to those who need them. For these reasons, using a One Health approach with pet owners experiencing homelessness is key to more effectively meeting this group’s needs and helping them to exit homelessness.

## 3. Multilevel Intervention Framework

The support needs of pet owners experiencing homelessness, whether they are staying in emergency shelters, sleeping on the streets or in vehicles, doubled-up with friends and family, or living in another place that is not their own, are not sufficiently addressed by existing homeless, housing, and healthcare systems. Though change is needed in service settings to better support this group, systemic and structural barriers related to housing policy and stigma must also be addressed. Informed by a One Health approach, we propose a multilevel framework for improving the health and well-being of people experiencing homelessness with pets (see Table 1; relevant definitions are provided in the text when interventions are discussed in detail). 

The framework’s policy, public, and service interventions are drawn from empirical research on pet ownership and homelessness, as well as the diverse backgrounds of the authors, which includes lived experience and professional practice with pet owners experiencing homelessness. Policy and service examples are integrated throughout the framework. Although these are mostly North American examples, the framework is not intended to be limited to this region and likely has applicability to homeless, housing, and healthcare systems in other countries around the world. Furthermore, the interventions that comprise the framework are intended to be beneficial to the homeless population broadly, but considerations for youth are also provided given this group’s unique needs.

The multilevel intervention framework provides policymakers; housing, human and animal health, and social service organizations; and homeless advocates with a set of actions that can be taken to more meaningfully support pet owners and augment efforts to prevent and end homelessness. Many of the proposed interventions are implementable within broader homelessness policy and strategic action plans. Although all interventions are expected to be beneficial for pet owners experiencing homelessness, leverage points that could facilitate implementation of the framework are likely to differ between communities. As such, intervention prioritization and implementation needs to be tailored regionally to the needs of people experiencing homelessness, the housing sector, and health and social service systems. 

### 3.1. Policy-Level Interventions

Aligned with a health equity approach, actions are needed at the policy level to remove barriers to accessing housing and emergency shelter for pet owners experiencing homelessness. Beginning with the latter, emergency shelter policies that prohibit pets are a prominent barrier to accessing temporary accommodation and accompanying health and social services [25,32]. Because of this, pet owners are more likely to sleep outdoors, in vehicles, or in derelict buildings [25,28]; living arrangements that are associated with higher mortality rates than sheltered homelessness [20]. More pet-friendly emergency shelters are needed to promote the health and well-being of people experiencing homelessness and their pets. Given estimates of pet ownership among the homeless population [25,26], social service systems should aim to have a minimum of 10% of emergency shelter beds be pet-friendly. 

There are examples of pet-friendly emergency shelters in many cities, which demonstrate that it is feasible to concurrently support people experiencing homelessness and their companion animals [26]. This practice is referred to as co-sheltering and can take different forms, including having pets sleep in the same room as their owners, having a designated onsite animal housing area, or partnering with animal health and welfare organizations (e.g., animal shelters, pet boarding services, animal rescues and fostering) to offer temporary boarding options. Wherever possible, it is recommended that pets stay with their owners so as to preserve the human-animal bond and reduce fears around pet loss [26]. It is also critically important to address known barriers, including organizational concerns, to the development of more pet-friendly emergency shelters. In a survey of emergency shelter service providers where pets were not allowed, common reasons for why this was included: health and safety of service users and providers, hygiene, space, potential damage to facility, noise, and cost [42]. Many of these issues can be prevented or mitigated through planning and partnerships. For example, implementation of screening procedures when pet owners enter the shelter, development of rules for having pets stay with people in the shelter, and communication of the responsibilities that pet owners have while in the shelter can be helpful in reducing potential health and safety issues. Furthermore, collaboration between the homeless service and animal health and welfare sectors can be mutually beneficial in developing and addressing issues related to pet-friendly emergency shelters. Animal health and welfare organizations can provide support and guidance around animal health and safety, co-shelter design, and access to veterinary services [26,61]. In turn, the animal health and welfare sector can benefit from a decreased likelihood that people experiencing homelessness need or are forced to surrender their pets, as well as increased access to an underserved population of pet owners [26].

Enacting more pet-friendly policies in emergency shelters will also help to create more fully low-barrier service options (i.e., programs that accept people how they are by having minimal requirements for accessing support), which are instrumental for accommodating the diverse and complex support needs of the homeless population. Accordingly, communities that are intending to increase their low-barrier emergency shelter options should explicitly include pet access requirements in requests for proposals. Some governments in the U.S. have begun doing this by incorporating pet-friendly shelter models into their broader strategies for reducing and ending homelessness. For example, in New York City, the Department of Homeless Services actively solicits innovative emergency shelter models to reduce unsheltered homelessness, including models that accommodate pets [62]. Similarly, some municipalities in California recognize that low-barrier shelters that accept pets, partners, and possessions are critical to their efforts to reduce unsheltered homelessness [63]. 

A similar problem exists in rental housing markets, with the lack of pet-friendly options perpetuating homelessness for people with pets. In a survey of over 100 landlords in the U.S., less than 10% stated that they allowed pets without restrictions, whereas 44% limited pets (by size and/or type) and 47% forbid them [64]. The barrier that this presents for pet owners experiencing homelessness is further exacerbated by widespread affordable housing shortages in most cities [65,66]. With low vacancies and no laws against ‘no pets’ housing policies, pet owners experiencing homelessness are left with few to no affordable housing options. Accordingly, there is an urgent need to remove ‘no pets’ policies in the rental housing market. Such a policy shift would be well aligned with the push toward housing as a human right [67], which recognizes that everyone has the right to adequate housing. From this perspective, housing rights cannot be disregarded due to pet ownership. The removal of ‘no pets’ policies in the rental housing market must also occur with a concurrent enforcement of those new laws. For example, in Ontario, Canada, it is illegal for landlords to reject housing applications or evict tenants on the basis of pet ownership. However, the law is poorly enforced, with many people also being unaware of their rights in this area [41,68]. As such, it is not uncommon for housing rentals to continue to be advertised as not allowing pets, deterring pet owners from pursuing the unit given the potential hassle and stress associated with dealing with an unaccepting landlord.

Although increasing pet-friendly affordable rental housing is expected to help pet owners to exit homelessness, such policy changes also have implications for maintaining tenancies. Pet owners’ security of housing tenure is strengthened, as they are not breaching tenancy terms by having pets and living with the constant threat of eviction [68]. Furthermore, as it is, pet owners may stay in unsatisfactory or poor quality housing due to challenges in finding other affordable pet-friendly housing [68,69]. Similarly, youth and women with pets may delay leaving unsafe home environments, including abusive relationships, due to limited pet-friendly housing options [28,70], thereby forcing them to choose between ongoing interpersonal violence, life without their companion animals, and/or homelessness. For these reasons, increasing the supply of affordable rental housing that does not prohibit pets has the potential to enhance housing stability and prevent homelessness.

Financial deposits required of pet owners, which are also known as pet deposits or pet bonds, represent another barrier in the rental housing market. In a survey of over 100 landlords in the U.S., 73% of those that offered pet-friendly housing required a pet deposit, which averaged about 40–85% of the monthly rent [64]. This was in addition to pet-friendly housing having higher rents than housing that did not allow pets. With prospective renters who have pets already feeling powerless and discriminated against in housing negotiations with landlords [69], pet deposit policies further consolidate the power held by landlords. For example, we have seen in our own work and experiences that although some pet deposits are refundable based on whether or not the animals cause any damage to property, such assessments are left to the discretion of landlords, often with little recourse for tenants. For people experiencing homelessness who have pets and are reliant on income supports to obtain housing, pet deposits are an additional hurdle that may be financially unsurmountable. Accordingly, amending rental housing laws to limit how much tenants can be charged as a security deposit (i.e., one month’s rent) and ban any additional non-service charges, such as non-refundable pet-related fees, would protect prospective tenants who have pets, especially those experiencing homelessness. The policy changes would contribute to parity and transparency in how issues related to pet ownership are rectified and could also be helpful for decreasing discrimination toward pet owners experiencing homelessness during the housing application process. Removal of non-service charges does not absolve pet owners of the responsibility for their companion animals. For example, tenants would still be responsible for any property damages caused by their pets and landlords may require that companion animals be spayed/neutered and up-to-date on vaccinations.

Housing First is a supported housing intervention that is effective in stably housing people experiencing homelessness who have complex support needs [71,72]. The intervention provides (a) a rent subsidy that can be used to immediately acquire market rental housing and (b) accompanying community-based mental health supports [73]. Tenants are not required to demonstrate housing readiness or commit to treatment to receive services. Moreover, housing and clinical services are provided separately, allowing tenants to retain supports if they move or lose their housing. As a result of the intervention’s strong research base, some jurisdictions in North America and Europe have adopted the Housing First philosophy as an evidence-based policy approach [73,74,75,76]. 

Implementation of a Housing First approach to housing policy has implications for pet owners experiencing homelessness. The impacts of pet ownership on outcomes in Housing First have not been fully investigated, though pets have been linked to greater community integration [77,78] and lack of integration remains an issue for Housing First tenants [79,80,81]. The intervention’s approach is also highly congruent with the needs of pet owners. With its low-barrier and person-centered principles, Housing First does not force people experiencing homelessness to choose between housing and their pets. Instead, Housing First practitioners are able to work with pet owners to find appropriate housing and advocate with landlords [82]. In this way, Housing First promotes the freedom of people experiencing homelessness to have and keep pets in their lives [83], which is a necessity to actualizing full individual choice—a central tenet of the intervention. Barriers may still arise from no-pet policies in the rental housing market; however, given the effectiveness of Housing First in stably housing people experiencing homelessness and its compatibility with the needs of pet owners, widespread implementation of Housing First in social policy will be beneficial to reducing barriers to pet owners exiting homelessness.

There are many financial costs to owning pets, not the least of which is veterinary care. Although there are examples of innovative veterinary programs that provide affordable services to people experiencing homelessness in some urban settings [26], such models of care are few and far between. Accordingly, veterinary services are financially inaccessible to many people who are low-income, including those experiencing homelessness [31,84]. To better support pet owners experiencing homelessness, there is a critical need for greater inclusion of veterinary medicine to provide basic pet care in specialized health services for the homeless population (e.g., street outreach, community health centers, and inner city health programs). One method for inclusion is having veterinary medicine be a funded position in these services. This occurred recently in California where one-time grants were made available through the Pet Assistance and Support Program to emergency shelters looking to provide accessible veterinary services, in addition to other animal-related supports (e.g., shelter and food), for pet owners experiencing homelessness [85]. Delivering veterinary care within health services that are sensitive to the needs of people experiencing homelessness may also help strengthen service connections and reduce pet owners’ fears that animals will be removed from them—a barrier to accessing veterinary care for the homeless population [31]. A second policy approach for increasing access to veterinary services for low-income pet owners is the inclusion of veterinary medicine in income support programs as a limited available benefit, allowing veterinarians to bill income support programs for provided services. To our knowledge, there are no known examples of this occurring in North America; however, such an approach would enable access to veterinary care beyond cities or regions where specialized, affordable services exist. Benefits for veterinary care would undoubtedly be insufficient for covering all expenses, though could offset the costs of treating some injuries and illnesses, including zoonotic diseases, and providing preventive care, such as immunizations and spay/neuter services, which can be important for obtaining and maintaining housing.

### 3.2. Public-Level Interventions

Pets can be a trigger for condemnation and harassment of people experiencing homelessness by members of the public. This is partially attributable to the high rates of public disapproval of pet ownership by people experiencing homelessness; approximately 25% in one study and 50% in another [47,86]. Common sentiments in Irvine’s seminal study for why the public oppose pet ownership by people experiencing homelessness, include: “They should not have a pet if they can’t take care of themselves,” “They can’t take care of the pet,” and “They shouldn’t have a pet if they don’t have a home” [47]. Drawing from practical experience, there is also a prominent discourse that “Having a pet is a privilege and not a right” and “If you can’t afford a pet, you shouldn’t have one.” Views that involve animal well-being concerns are unsupported by evidence. A study examining the health of 50 dogs owned by people experiencing homelessness found that they were no less healthy than 50 dogs owned by people with housing [87]. Furthermore, the dogs of people experiencing homelessness were less likely to be obese or have behavioral issues, such as aggression toward strangers and separation anxiety. There is also ample evidence that the pets of people experiencing homelessness are sufficiently fed, with many pet owners putting the needs of their pets above their own [28,44,47,87,88]. Beyond the concern for animal well-being, public disapproval of pet ownership by people experiencing homelessness results from stigmatization and discrimination. Accordingly, there is a need to enhance public knowledge and reduce stigma about the relationship between homelessness and pet ownership.

Although pet ownership and homelessness is a niche area of research, the issue has attracted considerably more attention in the media and public discourse. For example, two recent reviews identified fewer than 20 relevant studies on the subject [31,32], whereas searches of “homeless pet owners” on Google News and Twitter (within last year only) yielded more than 100,000 and 150 hits, respectively. The internet search results would suggest that the issue has an attentive public audience that could be potentially leveraged in public awareness initiatives aimed at dispelling myths associated with pet ownership and homelessness. Although no interventions aimed at the public are known to have been developed on this issue, structural stigma reduction programs on mental illness and HIV have yielded positive results and offer approach considerations [89]. 

Key educational messages in any public awareness initiative would need to address the primary criticisms of pet ownership by people experiencing homelessness, such as the baseless belief that people experiencing homelessness are unable to care for their pets, while concurrently highlighting the psychological benefits of the human-animal bond. The latter is a necessary component, as it offers opportunities for relatable stories with which the public can connect. Instead of perceiving pet owners experiencing homelessness as an outgroup with whom they have nothing in common, the public can find commonality in the human-animal bond, which is not bound to one’s housing situation. Lastly, this type of educational work is well-positioned for more cross-sectoral collaboration with animal health and welfare organizations. Bringing in new stakeholder groups with expertise on animal health and well-being will not only be helpful for reaching wider audiences but also for serving as partners in the promotion of policy- and service-level changes to better support pet owners experiencing homelessness.

Panhandling, which is also known as street begging, can be a precarious activity for pet owners experiencing homelessness that frequently results in confrontations with members of the public [23,28,36]. Yet, panhandling is often one of only a few income-earning activities in which pet owners experiencing homelessness can engage without being separated from their pets [36]. Furthermore, concerns about animal exploitation, which can be a source of conflict in interactions with the public, are often unfounded. Pets may increase monetary earnings from panhandling; however, pets are the primary beneficiary, as donations are predominantly pet food [36,90]. Given the necessity of panhandling for pet owners experiencing homelessness, reducing stigma associated with this activity will be helpful for enhancing this group’s safety on the streets. 

Anti-stigma interventions aim to reduce stigma and discrimination of a group through the provision of education (to replace myths with factual information) and contact (to challenge prejudicial attitudes and biases through direct and indirect interactions with the targeted group) [91]. Interventions to increase acceptance of panhandling by pet owners experiencing homelessness would come during a period of changing public attitudes. In the U.S., there has been a shift in perceptions of panhandling over the past three decades, toward greater compassion [92]. Still, major barriers to panhandling acceptability remain [93,94]. Most notably, panhandling is illegal or involves stipulations on how, when, and where this activity can occur in many communities [94]. As exposure to people experiencing homelessness (i.e., the frequency that people see this group each week) is positively associated with panhandling donations [95], laws that prohibit panhandling, or criminalize homelessness more broadly, reduce contact between people experiencing homelessness and the public that could be beneficial for promoting empathy and compassion. 

There are several key myths that must be addressed in any public intervention to destigmatize and enhance public knowledge about panhandling. First, contrary to the public perception that people experiencing homelessness make large amounts of money from panhandling, the median monthly income is $300 [96]. The earnings are not transforming people’s lives but rather help them to make ends meet while living in survival mode. Second, although drugs and alcohol are a reported expense of panhandlers, the primary source of spending is food [96]. Furthermore, people who panhandle report greater food insecurity than those who do not [95], highlighting the absolute poverty in which this population lives. Third, just as pet owners experiencing homelessness protect themselves from the public’s verbal assaults by rejecting the values underlying those messages [47], interventions need to redefine what pet ownership means to people experiencing homelessness. Here again, it is important to use the research, which shows that pets are sufficiently healthy, pet owners prioritize the needs of their companion animals above their own, and the psychosocial benefits of pet ownership. With appreciation of this evidence, it can then be understood that panhandling donations are likely to flow to meet the needs of pets before the people who own them, preserving and strengthening the relationship between human and animal. 

### 3.3. Service-Level Interventions

There are a range of interventions that can be used at the direct service delivery level to better support pet owners experiencing homelessness. This includes shared actions that can be taken by the many providers who serve this group (e.g., healthcare providers, homeless service providers, housing providers, veterinarians, animal health and welfare organizations), as well as actions that are unique to various professionals’ specific roles. Some of the proposed interventions likely exceed service providers’ current capabilities and scopes of practice, thus, requiring additional or specialized training. This may not be feasible for all service providers in which case simply recognizing what interventions are needed will be essential for referring pet owners experiencing homelessness to professionals specializing in those areas of treatment and care.

#### 3.3.1. Shared Actions for All Service Providers

Many health and social service systems are complex and minimally integrated, making them challenging to navigate for people experiencing homelessness [97]. As pets can be an additional barrier to accessing needed supports, service providers can assist this group by being informed of the community services that accept pets and offer pet-related supports (e.g., pet food, veterinary services), and their eligibility criteria. This includes being knowledgeable about what the services offer, where they are located and how people can get there if travel is required, when they operate, and what are the wait times for access. It is also recommended that service providers follow-up with pet owners experiencing homelessness after they have used a recommended community service to evaluate if their needs—both human and animal—were met. The feedback also enables service providers to enhance their own awareness of the helpfulness and accessibility of other community services by learning from pet owners’ experiences. 

The close human-animal bonds that people experiencing homelessness develop with their pets has been reported to be a protective factor against suicide [34,47]. When pets die, this can trigger bereavement and grief that is complicated by the undervaluation of this loss within society [98]. In some instances, suicidal ideation may develop or be exacerbated by the death of a companion animal [99]. Although people experiencing homelessness are at significantly higher risk of suicide than the general population [100], the role of pet loss has not been studied. Nevertheless, service providers should be aware of the harm-to-self risks associated with pet loss (e.g., death, surrender, runaway, removal) among people experiencing homelessness. Asking about suicidal ideation and making referrals to counseling and crisis management services are two important actions that can be taken when people experiencing homelessness have lost or are about to lose a pet. As substance use can increase during periods of grief [101], service providers can also provide support through education on harm reduction and safer use strategies to prevent overdose. 

Strengths-based practice is a person-centered approach to care that service providers can use when working with people experiencing homelessness, which strongly aligns with the support needs of pet owners. In contrast to conventional approaches that focus on service users’ problems, deficits, and pathology, a strengths-based approach aims to find solutions that consider people’s strengths, hopes, and goals [102]. From this perspective, pets reflect a set of strengths in the people who own them, as companion animals can be sources of responsibility, structure and routine, resourcefulness, pride, and motivation. Furthermore, pet owners experiencing homelessness can develop and build self-care skills derived from caring for dependent animals. For example, people experiencing homelessness report making changes toward heathier behaviors, such as reducing substance use, to better care for their pets and prevent separation [31,32]. Accordingly, service providers can build the capacities of people experiencing homelessness with pets by recognizing and leveraging their strengths in treatment and care, including personal capabilities and skills developed as pet owners.

Advocacy is another important role that service providers can embrace to support pet owners experiencing homelessness. To be an effective advocate for this group, having an awareness of regional pet ownership laws is an essential prerequisite. This includes being knowledgeable about people’s rights related to their pets in housing and service settings, as well as the consequences of law violations due to pet ownership. For example, it is important to know whether or not tenants can be evicted from housing for having pets and, if so, how this process unfolds. Advocating for the needs of pet owners experiencing homelessness with other service providers is also essential. Given the power imbalance between service users and providers, people experiencing homelessness can feel like they are unable to effectively advocate for their needs at times, which can leave them feeling powerless [103,104]. Accordingly, service providers can be a key ally in these efforts by advocating with other community programs for the accommodation of pets—not only emotional support and service animals—in service delivery. Lastly, given that service providers who work with pet owners experiencing homelessness are at the forefront of multiple complex social problems—the homelessness and unaffordable housing crisis, inequitable access to mental health services, and the overdose epidemic—they are an indispensable voice in systems change advocacy. Each of these problems can affect pet owners experiencing homelessness and service providers can be supportive of this group’s unique needs, such as the importance of access to pet-friendly housing and shelter, in their calls for action.

#### 3.3.2. Unique Actions for Specific Service Providers

The provision of support using a One Health approach can yield unique leverage points for some service providers to improve the health of people experiencing homelessness and their pets. For veterinarians and human health service providers, this includes the provision of education and harm reduction strategies on environmental (second-hand) tobacco smoke exposure by pets. Tobacco use is prevalent among the homeless population, with smoking estimates ranging from 57–82% [105]. Yet, pets can be a motivator to quit smoking. In a study of 698 pet owners who smoked cigarettes, over one-quarter said information about the harms of environmental tobacco smoke would motivate them to quit smoking [106]. Practical evidence also demonstrates that community-based veterinary clinics for pet owners experiencing homelessness and housing instability are feasible spaces to provide information about the harms of environmental tobacco smoke exposure to pets [107]. Animal health check-ups and grooming appointments are opportunities to engage pet owners in discussions about the impacts of tobacco, including cancer risks from inhalation and ingestion, on animals’ health [108,109]. Using motivational enhancement strategies to promote harm reduction is an important action that can be taken here to support this group.

Mental health service providers can also support people experiencing homelessness by providing letters documenting their animals as emotional support animals. Emotional support animals are pets deemed to have therapeutic benefits to their owners who have a mental illness but are not designated service animals [110]. In the U.S., emotional support animals are protected in housing and shelter by the Fair Housing Act. This law enables people to request to keep their emotional support animals as a “reasonable accommodation” to any ‘no pets’ restrictions and requires housing providers to allow “reasonable accommodations” involving any assistance animal that has therapeutic benefits for its owner [111]. The United Nations’ *Convention on the Rights of Persons with Disabilities* defines “reasonable accommodation” as the provision of necessary and appropriate modifications, which do not cause disproportionate or undue burden, so that people with disabilities are able to exercise their human rights and fundamental freedoms [112]. However, understanding of the laws that protect emotional support animals as “reasonable accommodations” can vary between organizations [24] so educational and advocacy efforts are important for promoting proper implementation of policies at the service level.

Although pets can conceivably be misrepresented as emotional support animals to obtain “reasonable accommodations” related to housing and travel [110], this is less likely to be an issue among people experiencing homelessness. This is largely because rates of mental illness and trauma exposure are very high among the homeless population [8,14]. In addition, companion animals have an integral role in coping with the adversities of homelessness [31,32]. Hence, people’s choice to keep their animals during homelessness, despite the barriers that this decision yields, is a reflection of the meaning and importance of this relationship and the emotionally supportive benefits that people experiencing homelessness derive from it. For more information and guidelines on how mental health service providers can support people experiencing homelessness with documentation of emotional support animals, including what a sample letter of support looks like, see the recent review by Hoy-Gerlach and colleagues [110].

## 4. Considerations for Youth Experiencing Homelessness

Although each of the proposed policy, public, and service interventions are expected to be beneficial to pet owners experiencing homelessness across the lifespan, youth have unique needs related to their developmental stage, reasons for homelessness (e.g., family rejection, childhood abuse and trauma), and approach to help-seeking that necessitate considerations in service delivery models [113]. Drop-in centers, which provide basic services with minimal barriers, such as food, hygiene, and some healthcare, are youth’s preferred location to access services [114]. Drop-in centers are uniquely appealing to youth experiencing homelessness because of their low-barrier approach with few restrictions and regulations, and despite the simplicity of this service model, drop-in centers operate as key entry points linking youth to other supports to help them exit homelessness, including housing and job training [115,116]. Studies have found that youth referred to drop-in centers, rather than emergency shelters, reported more service linkages, less substance use, and better HIV-related outcomes [114,117]. 

Given the benefits of drop-in centers for youth experiencing homelessness, it is instrumental that these services are welcoming and accommodating to youth with pets. This is particularly important given the mental health, self-care, and motivational benefits of pet ownership for youth, as well as because offering support for pets can increase trust and rapport between youth and service providers [28,36,88,118]. Unfortunately, recent research found a marginally significant decrease in use of drop-in centers among youth experiencing homelessness with dogs [119], suggesting that changes are needed to ensure that this group can access these essential services. 

Youth experiencing homelessness are more likely to use drop-in centers if they are perceived as safe, trustworthy, nonjudgmental, and physically and emotionally accessible [115,120]. For youth with pets, physical accessibility involves providing secure indoor or outdoor spaces for animals, with water, shade, and shelter, depending on climate and season. Drop-in centers should also provide pet food and serve as hub locations to mobile veterinary services or other veterinary medicine collaborations that provide affordable services for pet-owning youth experiencing homelessness. Emotional accessibility is key to ensuring youth feel welcome in service settings [120]. Drop-in centers can promote emotional accessibility by publicizing that the center is a pet-friendly space and actively acknowledging the importance of pets in youth’s lives, including through staff training, public materials, and on-site educational resources. As many youth access drop-in centers via referrals from peers [119], it is beneficial that youth experiencing homelessness without pets also be knowledgeable about the services that have pet-friendly policies so that they are able to share this information with their networks. Emotional accessibility also includes the extent to which youth feel welcome within the neighborhoods where drop-in centers are located [120]. Targeted public education campaigns within these neighborhoods may help to promote positive interactions that are grounded in respect and kindness between community members and pet-owning youth experiencing homelessness. 

## 5. Challenges for Framework Implementation

The multilevel framework offers a roadmap for how homeless, housing, and health services can better support pet owners experiencing homelessness. However, its pursuit will not be without challenges. The proposed interventions include new and different approaches to care that will require financial investments. Nevertheless, these recommendations are not made on the basis of remedying superficial misfortunates faced by this group but rather as a means of keeping pet owners experiencing homelessness alive and helping them to find and maintain housing. The interventions also strongly align with the right to adequate housing and shelter; a right that has been largely denied to pet owners experiencing homelessness to date or provided conditionally (i.e., housing and shelter are available if a companion animal is surrendered). In short, although financial investments are needed to support intervention implementation, they would represent smart spending toward producing more universally accessible homeless, housing, and health service systems. 

A second challenge is the training needs associated with the interventions. Human health and social services have a long history of supporting people who own pets but rarely have education on animal health and behavior [121]. Conversely, veterinarians and other animal health professionals may have little experience working with people who have complex histories of homelessness, mental illness, substance use, and other health conditions. Accordingly, there are consequential training needs for organizations and service providers who work with pet owners experiencing homelessness. One Health core competency frameworks may be helpful for identifying the skills and knowledge that direct service providers need to more effectively work with this group [49,122]. Partnerships with animal health and welfare organizations will be essential to bridging some of these education gaps [26]. 

A final challenge to implementation of the framework is regional differences. Interventions to support pet owners experiencing homelessness must be sensitive to local laws and policies, housing markets, service systems, and cultures, while recognizing contextual variations are likely to yield different priorities, opportunities, and barriers for action. Building this work into regional coalitions aimed at ending homelessness may be helpful for strengthening local support on the issue and leveraging change. Still, although the framework offers ideas and recommendations for moving forward with the proposed interventions, it is not a how-to guide. Resources exist for implementing some interventions, such as co-sheltering emergency shelters [24,26,61] and Housing First [82], though others will require community-developed practices and innovation.

## 6. Conclusions

The fundamental unmet need for pet owners experiencing homelessness, as with the broader homeless population, is housing. Yet, this group often faces unique barriers to obtaining temporary shelter and permanent affordable housing due to the lack of pet-friendly policies. Problems for pet owners experiencing homelessness intensify from there, from confrontations with uninformed or discriminatory members of the public to the financial costs of caring for pets. Given the harsh reality in which pet owners experiencing homelessness live, interventions are needed to improve the health and well-being of this population. Informed by a One Health approach, which recognizes the interconnectedness of human, animal, and environmental health, a multidimensional framework was developed that aims to provide support based on pet owners’ needs as opposed to care that is conditional based on the company they keep. By targeting policies and service approaches that indirectly exacerbate the hardships experienced by low-income pet owners, the framework is a set of deliberate actions to prevent and reduce homelessness by a group that is too often overlooked or unaccommodated. As the proposed interventions span multiple sectors, it is not feasible that they be implemented by any single entity. Instead, the framework components are intended to be integrated into broader homelessness policy and strategic action plans, as well as service delivery approaches. Coupling the implementation of these interventions with ongoing research and evaluation is also needed to transition from emerging practices to promising and best practices for supporting people experiencing homelessness with pets [123].

## Figures and Tables

**Table 1 animals-10-01869-t001:** A Multilevel Intervention Framework for Supporting People Experiencing Homelessness with Pets.

Level	Interventions
Policy	Increased supply of pet-friendly emergency shelters to ensure that a minimum of 10% of emergency shelter beds are accessible to people with pets Requirements that requests for proposals for low-barrier emergency shelters explicitly include pet access requirements Removal of ‘no pets’ policies in rental market housing and enforcement of laws that prevent discrimination in housing on the basis of pet ownership Limits on security deposit charges to one month’s rent and bans on any additional non-service charges, such as non-refundable pet-related fees Widespread implementation of a Housing First approach Inclusion of veterinary medicine in specialized health services for people experiencing homelessness to provide basic pet care
Public	Development of interventions to dispel myths associated with pet ownership and homelessness Development of interventions to destigmatize panhandling by people experiencing homelessness, including pet owners
Service	*Shared Actions for All Service Providers* Connection to community services that are accessible to pet owners experiencing homelessness, including veterinary care Awareness of the harm-to-self risks, such as suicidal ideation and increased substance use, associated with pet loss Use of a strengths-based approach to identify and leverage capacities developed from pet ownership in treatment and care Advocacy for the diverse needs of pet owners experiencing homelessness, including with other service providers and in systems change *Unique Actions for Specific Service Providers* Provision of education and harm reduction strategies to pet owners on environmental tobacco smoke exposure by animals (veterinarians and human health service providers) Provision of letters documenting animals as emotional support animals (mental health service providers)

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
