# Peer review of "A Multilevel Intervention Framework for Supporting People Experiencing Homelessness with Pets"

_animals, 2020, doi:10.3390/ani10101869_

Round 1

Reviewer 1 Report

I found the topic to be of great significance. The pets of homeless are generally ignored. While I agree on the issue of housing and removing no pets rules along with pet deposits, I believe that landlords should require all pets to be up to date on shots and spayed or neutered.

I work on behalf of homeless cats and a major contributing factor is absentee landlords who fail to require that pets are altered. There are so many low/no cost clinics that the cost is not an issue. What is an issue are the homeless cats left to survive on the streets. I am not suggesting that the homeless would abandon their pets but others who rent are a major cause of feline homelessness. Therefore, resolving feline homelessness requires landlords to require pets to be spayed/neutered. 

Other than that a wonderful study that addresses a widespread need for attention to the plight of the homeless and their pets.

Reviewer 2 Report

Very well written and structured article. Authors provide many insightful opinions and evidences that may help support people experiencing homelessness with pets. However, authors must work on the references throughout. As many arguments are lacking references and there are many reference errors.

Other comments:

Line 55: There are two reference [5]. Please fix it.

Line: 57-58: ‘have problematic substance use’. The meaning is unclear, please clarify it.

Line 64-65: ‘[Error! Reference source not found.,Error! Reference source not found.,Error! Reference source not found.]’ Please fix it.

Line 71: miss a comma. Please fix it.

Line 127-128: ‘there is clear evidence that pets have double-edged health effects on their owners’. The meaning is unclear, please clarify it.

Line 133-135: ‘With little accommodation of pets in service delivery to people experiencing homelessness, the contributions of companion animals to the health and well-being of their owners are either overlooked or de facto viewed to be inconsequential’. The logic of this sentence seems awkward. Please rephrase it.

Line 141-145: ‘By attending to the connection between human and animal health, a One Health approach would require pet-related supports be accessible to those who need them. For these reasons, applying a One Health approach to intervention for pet owners experiencing homelessness is key to more effectively meeting this group’s needs and helping them to exit homelessness.’

This is a good point. Would it be possible to provide a more specific example to elaborate this idea? For instance, what types of veterinary services are more commonly needed, barriers for seeking veterinary supports (e.g. cost, knowledge) and how such services/supports can help them exit homelessness?

Line 155-157: ‘Of note, research gaps and needs are omitted from the framework, as these have been identified in other recent work [32].’

For a better understanding, please briefly summarise or outline the research gaps that this framework is aiming to bridge, and then cite the reference.

Table 1: The table is a bit confusing. Which intervention belongs to which level? Please edit the table or provide more explanations in the footnote.

Line 162-154: ‘Beginning with the former, emergency shelter policies that prohibit pets are a prominent barrier to accessing temporary accommodation and accompanying health and social services.’ Please provide the reference(s) again.

Line 168-170: ‘Given estimates of pet ownership among the homeless population, social service systems should aim to have a minimum of 10% of emergency shelter beds be pet friendly.’ Please provide the reference(s) again that refer(s) to the number of 10%.

Line 171-172: ‘There are examples of pet-friendly emergency shelters in many cities, which demonstrate that it is feasible to concurrently support people experiencing homelessness and their animals.’ Please provide the reference(s).

Line 213-214: ‘…human right, which recognizes that everyone has the right to adequate housing’. Please provide the reference(s).

Line 239-241: ‘although some pet deposits are refundable based on whether or not the animals cause any damage to property, such assessments are left to the discretion of landlords, often with little recourse for tenants.’ Please provide the reference(s).

Line 272: Please change ‘this approach’ into some more specific descriptions. Not sure whether you are indicating Housing first approach, removal of policies being unfriendly to pets, or both.

Line 474-475, 478-479, 481, 486, 488, 491-492, 497, 501-502 and 505-506: ‘[Error! Reference source not found.,Error.]’ Please fix it.

Line 512-513: ‘Foremost, the proposed interventions include new and different approaches to care that will require financial investments.’ It will be useful to provide some evidences or examples that how these interventions may cost.

Reference

Please fix the reference format.

Reviewer 3 Report

Thank you for an important, well structures and practical article on the issue of pets and homelessness. This is an area within the support system for people experiencing homelessness that you rightly point out is under acknowledged, despite the importance of pets to owners and broad recognition of the benefits of pet ownership for people from all walks of life! Thank you also for referencing the potential links that can be made between pet ownership, homeless people’s needs, other sectors and also end homelessness movements!

Thank you also for calling people pet owners first, and people experiencing homelessness second!

I enjoyed reading this article and the journey it took me on very much. I work in the homelessness research space as a social scientist … and know there is some research emerging in this area too in Australia, which might be worth watching out for (watch for forthcoming research by Stone et al for the Australian Housing and Urban Research Institute on companion animals and housing pathways). We also have emerging data around ownership among rough sleepers, and evidence of low(er) barrier shelter options where housing pets is working. We also have a piecemeal systems of supports for pet owners experiencing homelessness, driven very much by relationships between and among certain vets and specialist homelessness services + foodbanks (for pet food) and the like. The landscape is shifting…slowly…

My detailed commentary on the paper is in comments on the pdf version attached. I also offer/reiterate the following:

The paper needs a few minor edits/additions to make it stronger and readable for as broader audience as possible (and needed to get the message out!).

Some further clarity around the method, per my commentary in the paper would demonstrate the robustness of the paper, i.e. you mentioned lived experience involvement, but how? A short couple of sentences on method of development would be really effective.

Definitions need thinking through. Especially pet v companion animal v service animal v support animal. I know this is somewhat jurisdiction/context specific, but general definitions could be provided, in a table perhaps? People definitely get assistance, companion and service animals confused. And I’m not sure everyone understands that pets are companion animals. That has certainly been our experience in the housing sector generally. And, thinking of the specialist homelessness sector and beyond veterinary medicine audience, best not to assume understanding.

Other key definitions are adequately provided in the paper, thank you (Housing First, One Health, low barrier, panhandling, strengths-based). I did note on my first read of the paper that there was assumed understanding of these terms in the presented multilevel framework, but the detail needed is there, its just a bit later in the paper. Can you please add a sentence at the end of the first paragraph in section 3 (line 157) to explain that definitions are provided later in the text of the paper. Some of the concepts identified in the framework will simply not be known to the journal audience: veterinary, other medical, homelessness sector or other social services. (Panhandling in particular is not a term that translates to some contexts, Australian for example. We call it street begging).

I think there needs to a bit of work throughout the paper on three other things:

  1. Is the framework targeted to specialist application only (i.e. via community vet outreach services/vets within services) or for mainstream application, i.e. all vets? I’m not sure I get a clear enough picture here. Can some words be added around this? There is one minor sentence about broad implementation, which needs teasing out more.

In my jurisdiction, some vet services are provided for homeless people (including people sleeping rough) but they are single/groups of vets providing in-kind services based on certain days (including one off event days) within service premises (no outreach here yet, although that would be fabulous for our rough sleeping population!). These vet services for homeless people are not at all government supported, and the vets wouldn’t necessarily have specialist training in engaging with people experiencing homelessness. If this is about ‘mainstreaming’/embedding support for pet owners who are homeless (and I hope it is) there is definitely multiple layers of training to be provided; for vets and for specialist homelessness services and interfacing providers (health, mental health, social supports etc).

  1. Some further attention needs to be paid to the use of the term homeless. Can you define who/what you mean a little more clearly, even if it is we mean ‘homelessness’ in the broadest sense here… capturing people street sleeping/rough sleeping, people doubled up (we call couch surfing, you might call sofa surfing?), people who are chronically homeless, people who cycle in and out of homelessness and people escaping violent situations. (The framework will also have value in prevention work for people at risk of first time or recurrent homelessness; you could say this too!). You rightly point out the experience of pet ownership and supports for youth are different. They are also different for people sleeping rough and for people experiencing violence impacting safety at home. I read much of the paper in the frame of people sleeping rough. Is that how you intend it to be read? This just needs a few more words around the diversity of homelessness types and experiences and how this shapes people’s experiences traversing systems with pets.
  2. Framing the geographical context of the paper, I suggest labelling it more North American in focus given where you are all from, but clearly stating the application beyond North America.

Across the whole paper there needs to be a fix of some of the cross-references, word/PDF has done its usual and lost some!

A final proofing and reference check would also be good.

This is a worthy practical contribution to the literature. Thank you for contributing it!
